# A Chitosan-Based Fluorescent Probe Combined with Smartphone Technology for the Detection of Hypochlorite in Pure Water

**DOI:** 10.3390/molecules28176316

**Published:** 2023-08-29

**Authors:** Xushuo Yuan, Wenli Zhang, Li Liu, Yanfei Lin, Linkun Xie, Xijuan Chai, Kaimeng Xu, Guanben Du, Lianpeng Zhang

**Affiliations:** 1Yunnan Provincial Key Laboratory of Wood Adhesives and Glued Products, Southwest Forestry University, Kunming 650224, China; yuanxushuo99@163.com (X.Y.); m13304792956@163.com (W.Z.); liuli0573@126.com (L.L.); xielinkun@163.com (L.X.); xjchai@126.com (X.C.); xukm007@163.com (K.X.); guanben@swfu.edu.cn (G.D.); 2College of Biological, Chemical Sciences and Engineering, Jiaxing University, Jiaxing 314001, China

**Keywords:** chitosan, fluorescence, sodium hypochlorite, smartphone technology

## Abstract

Using chitosan as a raw material, 1,8-naphthimide as the fluorescent chromophore, and sulfur-containing compounds as the recognition groups, a novel naphthimide-functionalized chitosan probe, CS-BNS, for the detection of ClO^−^ was successfully synthesized. The modification of chitosan was verified by SEM, XRD, FTIR, mapping, ^13^C-NMR, TG and the structure of the probe molecule was characterized. The identification performance of the probes was studied using UV and fluorescence spectrophotometers. The results show that CS-BNS exhibits a specific response to ClO^−^ based on the oxidative reaction of ClO^−^ to the recognition motifs, as well as a good resistance to interference. And the probe has high sensitivity and fast response time, and can complete the detection of ClO^−^ in a pure water system within 60 s. The probe can also quantify ClO^−^ (y = 30.698x + 532.37, R^2^ = 0.9833) with a detection limit as low as 0.27 μM. In addition, the combination of the probe with smartphone technology enables the visualization and real-time monitoring of ClO^−^. Moreover, an identification system for ClO^−^ was established by combining the probe with smartphone technology, which realized the visualization and real-time monitoring of ClO^−^.

## 1. Introduction

Hypochlorous acid (HClO) and sodium hypochlorite (NaClO) are widely present in living organisms and the environment. HClO/ClO^−^ has a strong oxidizing and bleaching effect, and is often used as a bleaching agent and disinfectant for water treatment. It is widely used for sterilization and disinfection of drinking water, tap water, swimming pools, hospitals and other places [1,2]. However, the presence of excessive residual chlorine in domestic water poses a certain threat to human health. For example, it can cause asthma, esophagitis, laryngitis and spontaneous vomiting. HClO can protect the health of the organism by killing pathogens and preventing their invasion, playing a crucial protective role in the immune system [3,4]. However, excess HClO/ClO^−^ in the living body can lead to severe damage to the basic biological components such as proteins, carbohydrates and lipids. This leads to the risk of many diseases such as Alzheimer’s disease, kidney disease, neurodegenerative diseases, diabetes, cardiovascular disease, rheumatoid arthritis and cancer [5,6,7,8,9,10]. Therefore, it is important to develop a detection method for HClO that can be easily realized in real water samples.

Fluorescent probes have the advantages of easy synthesis, simple operation, high sensitivity, high selectivity and short response time, and are widely used in biology and environmental monitoring fields. In general, fluorescent probes consist of two parts: a fluorescent group and a recognition group [11,12,13,14,15,16]. Fluorescent probes based on different fluorophores, recognition groups, and detection mechanisms have received much attention in the detection of HClO [17,18,19]. Hu et al. synthesized a chitosan-based fluorescent probe, SWFU, based on the property that the methylthio group is easily oxidized to sulfoxide by ClO^−^. The detection limit of the probe was as low as 1.4 μM. The fluorescence color changed from yellow to blue, and the presence of ClO^−^ could be determined by the naked eye [20]. Song et al. synthesized a novel probe, HCA-Green, which uses 4-bromo-1,8-naphthalimide as a fluorophore and p-aminophenol as a recognition group. When HClO was added, the phenol group was cut off, thereby disrupting the PET effect and exhibiting a significant fluorescence intensity enhancement [21]. Yin et al. designed and synthesized a fluorescent probe based on fluorescein and containing a catechol structure, which has a spiro-loop structure and does not fluoresce by itself. The reaction with ClO^−^ opens the ring and shows an intense yellow-green fluorescence. The quantum yield of the probe reacting with NaClO (Φ = 0.87) was 1240 times higher than that before the reaction (Φ = 0.0007), which can effectively detect HClO in living cells [22]. Duan et al. designed and synthesized a BODIPY ratiometric fluorescent probe. The reaction mechanism is oxidation of the thioether group to sulfoxide by HClO, and the detection limit of the probe is as low as 59 nM with a response time of <30 s. The probe has been successfully applied to image endogenous and exogenous ClO^−^ in zebrafish and mice [23]. Chen et al. developed a two-color fluorescent probe based on the association of methylene blue and naphthalimide. The probe was successfully applied for the qualitative and quantitative detection of HClO and H_2_S in vitro and in vivo [24]. Zhang et al. introduced the Schiff base structure into the bisquinolone unit and designed and synthesized two new efficient fluorescent probes (DQNS, DQNS1). The probes can be used to detect Al^3+^ and ClO^−^ in real water samples and live cells [25].

Chitosan is a safe, non-toxic, low cost, degradable and high performance natural polymer. Its reserves are abundant, second only to cellulose [26,27,28]. The extraction route is wide, and chitin is extracted from the cell wall of shrimp, crab, insect, or bacteria and algae, and then obtained by removing part of the acetyl group. The presence of a large number of functional groups such as –OH and –NH_2_ in the molecular structure of chitosan makes it easy to modify [29]. Hydrolysis, gelation, cross-linking, grafting, redox and complexation reactions can occur to form materials with a wider range of applications [30,31,32], including agriculture, industry, environmental monitoring and nanomedicine [33,34]. Based on these advantages, research on chitosan-based fluorescent probes has been continuously explored by scholars both at home and abroad.

The 1,8-naphthalimide structure has strong fluorescence emission, better photostability and has been widely used in the synthesis of fluorescent probes [35,36]. Based on the susceptibility of sixth main group elements (e.g., S, Se, Te) to oxidation by ClO^−^, researchers have developed a number of fluorescent probes with excellent performance. Choi and Kim et al. synthesized fluorescent probes for ClO^−^ detection using BODIPY as fluorophore and introducing methyl phenyl sulfide as a recognition group in the structure. A significant fluorescence burst occurred after the recognition of ClO^−^ by the probe molecule due to the photo induced electron transfer (PET) process. The probe has the advantages of large Stokes shift and higher quantum yield [37]. Liu et al. synthesized BODIPY-based fluorescent probe 1. After the recognition of HClO, the probe promoted the oxidation of methyl phenyl sulfide to form a sulfur–oxygen double bond, which inhibited the PET process from methyl phenyl sulfide to BODIPY. The detection limit of probe 1 is as low as 23.7 nmol/L, which enables highly sensitive and selective detection of hypochlorous acid [38].

In this study, a sulfur-containing fluorescent probe, CS-BNS, was designed by introducing naphthylimide as a fluorescent moiety in the chitosan skeleton as shown in Figure 1. This probe senses ClO^−^ by an oxidation reaction mechanism. Structural and fluorescence characterization of CS-BNS were performed to further evaluate its recognition properties for ClO^−^. The probe can be identified by the “naked eye” due to its distinct color change from colorless to yellow. To clarify the effect of CS-BNS on identifying slight color changes between different ClO^−^ concentrations, the probe solution was combined with smartphone technology. Using color recognition software for a smartphone, RGB values of probe solutions are identified when different ClO^−^ concentrations are added. A system was constructed for the recognition of ClO^−^ by the probe.

## 2. Results and Discussion

### 2.1. Structural Characterization of the CS-BNS Probe

#### 2.1.1. SEM Analysis

To verify the successful synthesis of the fluorescent probes in this study, a series of characterization analyses of the chitosan derivatives during the synthesis were performed. Figure 2 shows the microscopic morphology of CS, CS-B and CS-BNS magnified to 5 μm versus 500 nm. The images show that the untreated chitosan feedstock has a regular surface texture with a dense structure. After introduction of the fluorescent group naphthylimide, the surface of the compound showed a fine rod-like structure and the dense structure became looser. And the compounds were seen to be loaded with granular substances on the surface. This indicates that a series of modifications changed the external and internal morphology of chitosan. The CS-BNS probe was synthesized after further introduction of recognition groups, and the structural surfaces of the compounds showed large voids and pores with a regular 3D porous structure. These meshes and pores facilitate the recognition of target ions by the CS-BNS probe [39].

#### 2.1.2. XRD Analysis

The XRD patterns of CS and CS-BNS are shown in Figure 3. It can be seen that the positions of the diffraction peaks of chitosan and CS-BNS basically remain the same, with the main diffraction peaks at around 12.59° and 20.00°. However, compared with chitosan, CS-BNS showed a new diffraction peak at 26.12°. This may be caused by the introduction of fluorescent groups with recognition groups. We further analyzed this by FTIR spectroscopy.

#### 2.1.3. FTIR Analysis

The FTIR spectra of CS, CS-B and CS-BNS are shown in Figure 4. Among them, the peaks at 3322–3480 cm^−1^ are overlapping stretching vibrations of N–H and –OH. The peaks appearing at 1640 cm^−1^ and 1384 cm^−1^ are the C–N and C–C bonding vibrations, respectively [40]. The peak at 1030 cm^−1^ is formed by the stretching vibration of C–O–C [41]. All three curves have peaks at these locations, indicating that the chitosan derivatives maintain their basic structure before and after modification.

Compared to the spectrum of CS, CS-B shows new peaks at 1590, 1662 and 1725 cm^−1^, which are characteristic peaks of C=O [42]. In addition, the new peaks of CS-B at 1500 cm^−1^ are attributed to benzene ring skeleton stretching vibrations. This indicates that we successfully synthesized CS-B by introducing 4-bromo-1,8 naphthalic dicarboxylic anhydride on the chitosan backbone. The newly introduced functional group on CS-BNS could not be seen in the FTIR spectra compared to CS-B. Therefore, we further analyzed the probe structure with the help of mapping and ^13^C-NMR.

#### 2.1.4. Mapping Analysis

The synthesis process of the fluorescent probes was further confirmed by the elemental distribution of each compound in Figure 5. Chitosan is a polymer composed of C, N, O. The element Br was detected in compound CS-B, which is the characteristic element on the fluorescent group 4-bromo-1,8-naphthalic dicarboxylic anhydride. This indicates that the fluorescent group was successfully introduced into the molecular chain of chitosan. The presence of S element was detected in CS-BNS, but no Br element was detected. This indicates that after the recognition group was introduced, the brominated aromatic hydrocarbons in CS-B were successfully substituted with thiomorpholine. Figure 6 shows the distribution of C, N, O and S elements in CS-BNS. Among them, S elements are abundant and uniformly distributed, which provides more reaction sites for the probe to sense ClO^−^ through the oxidation reaction mechanism.

#### 2.1.5. ^13^C-NMR Analysis

The structures of CS and CS-BNS were analyzed in combination with ^13^C-NMR. As shown in Figure 7, the peaks in the yellow region are from the carbon on the chitosan backbone at C1–C6. In contrast to CS, the modified CS-BNS probe shows a series of new peaks in the purple region, which come from the carbon on the recognition group aromatic ring region C8–C18. The new peak at 160.14 ppm corresponds to the carbonyl C7, C17 of the recognition group. In addition, new peaks at 23.76 ppm, 45.13 ppm, correspond to C19, C20 on the recognition group, respectively. All of these results clearly demonstrate that chitosan was successfully modified.

#### 2.1.6. TG Analysis

Figure 8 shows the thermogravimetric analysis (TG) and micro-quotient thermogravimetric analysis (DTG) plots of CS, CS-B and CS-BNS. In the first stage (35–110 °C), the weight of all three decreased to varying degrees due to the evaporation of water from the samples. Among them, the thermal weight loss of the probe CS-BNS was 4.78%. Significant weight loss was observed in the second stage (200–400 °C). The DTG curves showed that the maximum degradation temperature of CS-BNS was 224.29 °C, higher than both CS and CS-B, indicating that the thermal stability of the probe was better.

### 2.2. Spectral Characterization of the Probe CS-BNS

#### 2.2.1. Concentration Gradient for ClO^−^

The tests were performed at room temperature. All fluorescence emission spectra were measured at an excitation wavelength of 336 nm, with the detection wavelength set at 350–650 nm and the slit set at 5 nm × 5 nm. The fluorescence response relationship of CS-BNS to different ClO^−^ concentrations is shown in Figure 9a. In the pure water system, the emission peak of CS-BNS appeared at 494 nm. The fluorescence intensity of the probe gradually increased with the increase of ClO^−^ concentration. Under natural light, there is no obvious color change of the probe solution after the addition of ClO^−^. Under 365 nm UV light, a bright yellow fluorescence of the solution can be observed by the naked eye.

As shown in Figure 9b, the fluorescence intensity was linear with different concentrations of ClO^−^ (0–50 μM). According to the formula for the limit of detection (LOD), LOD = 3σ/s, where σ is the standard deviation of the fluorescence intensity of the probe blank sample, and s is the slope obtained from the linear fit of ClO^−^ concentration–fluorescence intensity [43]. The limit of detection (LOD) was calculated to be 0.27 μM. The results of previous studies on ClO^−^ fluorescent probes in recent years are compiled in Table 1. The comparison shows that the response of the probe to ClO^−^ has good sensitivity and fast response time.

#### 2.2.2. Selectivity and Interference Resistance

We investigated the ability of CS-BNS to detect ClO^−^ in complex environmental samples. As shown in Figure 9c, the selectivity and anti-interference experiments of the probe for ClO^−^, Al^3+^, Ba^2+^, Ca^2+^, Cu^2+^, Fe^2+^, Mg^2+^, Na^+^, Fe^3+^, Ni^2+^, Ag^+^, H_2_O_2_, SO_3_^2−^, SO_4_^2−^, NO_2_^−^, NO_3_^−^ and OH^−^ were carried out. The results showed that only ClO^−^ caused a significant increase in the fluorescence intensity of CS-BNS at 494 nm. No insignificant changes were observed with the addition of other interferents. Thus, the probe can be used for selective detection of ClO^−^. Moreover, the recognition of ClO^−^ by the probe was not affected when ClO^−^ coexisted with other interferents. This indicates that CS-BNS has anti-interference ability and is less affected by interferents.

#### 2.2.3. Time Response to ClO^−^

To further investigate the response rate of CS-BNS to ClO^−^, the change in fluorescence intensity was detected every 30 s after the addition of ClO^−^ to the probe solution. As shown in Figure 9d, the results indicated that the fluorescence intensity increased sharply after the addition of ClO^−^. The fluorescence intensity rapidly reached the maximum value in 60 s and remained stable at 300 s. This indicates that the fluorescence intensity of the CS-BNS was not changed after the addition of ClO^−^ to the solution. This indicates that CS-BNS has excellent rapid detection ability. It can realize the real-time and rapid monitoring of ClO^−^.

#### 2.2.4. Effect of pH

Recognition of targets by probe molecules is highly dependent on factors such as the pH of the environment [47]. The effect on CS-BNS in the range of pH = 3–10 in the presence or absence of ClO^−^ was recorded, respectively, as shown in Figure 10. When ClO^−^ was not added, the fluorescence intensity of the probe CS-BNS at 494 nm did not change significantly in different pH environments, and all of them showed weaker fluorescence intensity. This indicates that the probe molecule has better environmental acid–base tolerance.

When ClO^−^ was added, the fluorescence intensity of the probe was slightly enhanced at pH = 3. The fluorescence intensity was significantly enhanced with the increase of pH where the fluorescence intensity at pH = 7 reached the maximum value. After that, the fluorescence intensity slightly decreased with further increase in pH. This result also demonstrated that the CS-BNS probe is stable over a wide pH range and has the ability to detect ClO^−^ under physiological conditions.

### 2.3. Smartphone-Based Detection of ClO^−^

Fluorescent probes are often detected and analyzed using a fluorescence spectrophotometer. This equipment is expensive and requires a certain level of expertise from the experimenter during its use. Therefore, we established a simple method for detecting ClO^−^, which is based on smartphone recognition software. As shown in Figure 11a, probe solutions containing different ClO^−^ concentrations (0–50 μM) were first configured and placed under a 365 nm UV lamp. The yellow fluorescence of the solution increased with the increase of ClO^−^ concentration. The RGB values were recorded separately by taking pictures using the “What a color?” software in a smartphone [48]. The fitted curve of R/B value versus ClO^−^ concentration was established as shown in Figure 11b.

Based on the equation, LOD = 3σ/s, the LOD of this smartphone detection technique for recognizing ClO^−^ by RGB was calculated to be 4.069 μM. The result is higher than the LOD for detecting ClO^−^ using a fluorescence spectrophotometer. This is due to the weak fluorescent color change when the ClO^−^ concentration is low, making it difficult for smartphone recognition software to distinguish the small color change in the solution. This shows that using a fluorescence spectrophotometer is a reliable method when micro-volume, fine detection is required. Compared with the fluorescence spectrophotometer detection method, the smartphone identification detection method has the advantages of being portable, easy to operate and low cost. It is suitable for some daily detection and real-time analysis.

## 3. Materials and Methods

### 3.1. Materials

Materials included CS (Chitosan: deacetylated 90%); 4-bromo-1,8-naphthalic anhydride (C_12_H_5_BrO_3_); acetonitrile (C_2_H_3_N); thiomorpholine (C_4_H_9_NS); and sodium hypochlorite (NaClO). Analytical purity-grade reagents were used in all experiments.

### 3.2. Experimental Process

The synthetic route of the CS-BNS probe and the mechanism of sensing ClO^−^ are shown in Figure 1. CS-BNS was synthesized by a two-step process using chitosan as a raw material. First, chitosan was reacted with 4-bromo-1,8-naphthalic anhydride in the presence of acetonitrile to synthesize CS-B. Then, the brominated aromatic hydrocarbon on CS-BS was substituted with thiomorpholine to synthesize the CS-BNS probe.

#### 3.2.1. Synthesis of CS-B

CS (0.554 g) was taken with 4-bromo-1,8-naphthalic anhydride (1.773 g) with acetonitrile as a solvent. The reaction mixture was placed in a magnetic stirrer and stirred thoroughly at 83 °C for 5 h. At the end of the reaction, the reaction mixture was placed in a vacuum rotary evaporator and the acetonitrile solution was evaporated. A light-yellow powder compound, CS-B, was obtained.

#### 3.2.2. Synthesis of CS-BNS

CS-B (1.1 g) was taken with thiomorpholine (0.12 g) using acetonitrile as a solvent. The reaction mixture was placed in a magnetic stirrer and stirred thoroughly at 83 °C for 6 h. The reaction mixture was then placed in a vacuum rotary evaporator and the acetonitrile solution was evaporated. Finally, a yellow-brown fluorescent probe powder, CS-BNS, was obtained by washing (acetonitrile and ethanol), dialysis (acetonitrile) and drying steps.

### 3.3. Properties of CS-BNS Probe

#### 3.3.1. Scanning Electron Microscope (SEM) and Energy Dispersive X-ray Spectrometer (EDS-Mapping) Analysis

The samples to be tested were placed on an aluminum grid and examined by field-emission scanning electron microscopy on a Nova Nano SEM 450 microscope (FEI, Hillsboro, OR, USA). At least five fields were examined for each sample. The distribution of elements in the samples to be tested was observed by EDS-Mapping.

#### 3.3.2. X-ray Diffraction (XRD) Analysis

The samples to be tested were examined by X-ray diffractometry on a Rigaku Ultima IV X-ray diffractometer (Rigaku Corp., Tokyo, Japan) (XRD, Ulti) using a scanning angle from 10° to 40°, a step size of 0.026° (accelerating current = 30 mA and voltage = 40 kV) and Cu-Kα radiation of λ = 0.154 nm.

#### 3.3.3. Fourier Transform Infrared Spectroscopy (FTIR) Analysis

The samples to be tested were mixed with KBr, pressed into a pellet and analyzed on a Nicolet i50 FTIR Analyzer (Thermo Scientific, Waltham, MA, USA) with a scanning range of 500 to 4000 cm^−1^ and 64 scans.

#### 3.3.4. Nuclear Magnetic Resonance (NMR) Analysis

The samples to be tested were examined using a Bruker AVANCE 600 spectrometer (Bruker Corporation, Billerica, MA, USA).

#### 3.3.5. Thermogravimetric (TG) Analysis

The sample were analysis on a TGA92 thermogravimetric analyzer (KEP Technologies EMEA, Caluire, France). N_2_ was used as the shielding gas and Al_2_O_3_ was used as the reference compound. The temperature was increased from 35 to 800 °C at a rate of 20 °C/min to generate a thermogravimetric curve.

#### 3.3.6. Ultraviolet Spectral Analysis

The samples to be tested were examined using a UV-vis spectrophotometer (UV-2600, Shimadzu, Kyoto, Japan).

#### 3.3.7. Fluorescence Spectral Analysis

The samples to be tested were examined using a fluorescence spectrophotometer (F-7100, Naka Works, Hitachi High-Tech Science, Ltd., Tokyo, Japan). The tests were performed at room temperature. All fluorescence emission spectra were measured at an excitation wavelength of 336 nm, with the detection wavelength set at 350–650 nm and the slit set at 5 nm × 5 nm. The CS-BNS test probe was tested for its concentration gradient, selectivity and immunity to interference, time response and the effect of varying pH.

### 3.4. Smartphone-Based Detection of ClO^−^

ClO^−^ (0–50 μM) was added to the probe solution. In combination with the “What a color?” color recognition software for smartphone, the RGB values of the probe solution at 365 nm were identified when different ClO^−^ concentrations are added. Differences in red (R), green (G) and blue (B) color intensities were obtained for each group of solutions [49]. The linear relationship between R/B value and ClO^−^ concentration was used to construct an identification system for ClO^−^ concentration.

## 4. Conclusions

In summary, we designed and synthesized a chitosan-based fluorescent probe, CS-BNS, which is more advantageous than common small molecule fluorescent probes for biomass resource utilization and environmental protection. CS-BNS enables the detection of ClO^−^ in pure water based on the fact that S readily undergoes oxidation reaction and triggers naphthylimide fluorophore after specific recognition with ClO^−^. This leads to the emission of yellow fluorescence visible to the naked eye, thus realizing the recognition of ClO^−^. The CS-BNS probe has high sensitivity, fast response time, good immunity to interference and stability over a wide pH range. The detection limit of the probe was as low as 0.27 μM and the response time was 60 s. This result was superior to the findings of other ClO^−^ fluorescent probes. In addition, we successfully combined the CS-BNS probe with a smartphone color recognition software to construct a system for identifying the concentration of ClO^−^. This demonstrates that CS-BNS has great potential for practical application in the concentration detection of real water samples. This easy and real-time smartphone detection further enhances the practical application possibility of fluorescent probes. Meanwhile, this study provides a reference for the utilization of chitosan and a new design route for the synthesis of fluorescent probes for the detection of ClO^−^.

## Data Availability

Data will be made available on request.

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
