# Peer review of "A Chitosan-Based Fluorescent Probe Combined with Smartphone Technology for the Detection of Hypochlorite in Pure Water"

_molecules, 2023, doi:10.3390/molecules28176316_

Round 1

Reviewer 1 Report

A chitosan-containing fluorescent sensor for detection of hypochlorite in water has been studied. The subject is interesting, applicable and valuable. However, there are some important points which should be addressed, clarified and/or discussed in the revised version. Therefore, I suggest major revision of the manuscript based on the following comments:

1.       In Figure 4, the kinds of elements in the elemental distribution images are not clear.

2.       The authors should comment on the stability of the sensor, especially after many cycles.

3.       The pH of pure water is 7 at 25C. This work was done at a different pH of 7.4, as mentioned in Figure 8. This means that the authors should discuss the possible effect of pH on the sensitivity. In fact, chemical absorption within nanoporous of electrodes/membranes strongly depends on the pH of the environment and the surface charge of the chemicals (see, for example, [Journal of Molecular Liquids 358 (2022) 119210]). This issue should be addressed and discussed in the revised version. 

4.       It has been mentioned that “Fluorescent probes are easy to synthesize, simple to operate, sensitive, selective, and have short response times”. This statement can be further supported by more updated references such as [Highly sensitive selective sensing of nickel ions using repeatable fluorescence quenching-emerging of the CdTe quantum dots]. All of the references of [5-7] are related to 13 years ago.  

5.       The limit of detection has been mentioned 0.27 micro molar. But, Figure 9a shows that the distinguished steps for monitoring the color change is 10 micro molar. Now, there is two order of magnitude difference between the smartphone technology result and detection limit. This issue should be clarified in the revised version. 

6.       It has been mentioned that “Hydrolysis, cross-linking, grafting, redox and complexation reactions can occur to form materials with a wider range of applications”. This can be further completed and supported as follows: “Hydrolysis, gelation, cross-linking, grafting, redox and complexation reactions can occur to form materials with a wider range of applications [J Tissue Sci Eng 2017, 8:3, 1000212]”.

7.       One of the important characteristics of sensors is their selectivity. Now, whether can this probe selectively sense the hypochlorite? This subject should be addressed and discussed in the revised version, using suitable supports.

8.       It has been stated that “This has led to a wide range of applications in agriculture, industry and environmental monitoring.”. This needs to be improved and supported as follows: “This has led to a wide range of applications in agriculture, industry, environmental monitoring and nanomedicine [Environmental Chemistry Letters volume 17, pages1667–1692 (2019)] & [Journal of Controlled Release 350 (2022) 175-192].”.

9.       In Figure 6, the vertical axes need labels and unit. The negative labels seem meaningless. 

Minor revisions can be considered. 

Reviewer 2 Report

The manuscript describes a slight extension of previous work on closely relatedchitosan-naphthalimide-based fluorescent probe. Nevertheless, detection can be done easily using a smartphone and, therefore, I recommend the manuscript for publication in Molecules upon some minor revisions:

1.) -1,8- (not 1, 8-)

2.) Each acronym should be defined the fist time it appears in the text. For example, chitosan (CH). It would make manusctipt much easier to read and understand.

3.) Scheme 1 and Sections 2.1.1. and 2.1.2.: The authors say, that the reaction mixture was heated at 100 °C. How is this possible, since acetonitrile (solvent) gas b.p. 83 °C? Please explain.

3.) Experimental procedures should be written in a manner, which is usual for description of synthetic procedures in the field of organic synthesis.

4.) Line 78: ......oxidized by ClO-, a sulfur.....

5.) line 89: .....anhydride; acetonitrile......

Round 2

Reviewer 1 Report

The authors tried to revise the manuscript. However, there are still some minor points which should be addressed before the final publication, as mentioned below:

Concerning Comment 3#, I understand that the subject of pH dependence can be considered for future works. But, preliminary mentioning this issue is helpful for the readers. I once again suggest introducing this point in the revised version based on literatures (please refer to the original comment).

Concerning Comment #4, the response is “We have revised the section and cited the latest literature.”. It is Ok. But, it seems that the article mentioned in the original comment is missed in the revised version.

After considering theses points the manuscript can be considered for publication.   

Some minor revisions can be considered. 
